# Multimodal Chain of Continuous Thought for Latent-Space Reasoning in Vision-Language Models

## Abstract

Many reasoning techniques for large multimodal models adapt language model approaches, such as Chain-of-Thought (CoT) prompting, which express reasoning as word sequences. While effective for text, these methods are suboptimal for multimodal contexts, struggling to align audio, visual, and textual information dynamically. To explore an alternative paradigm, we propose the Multimodal Chain of Continuous Thought (MCOUT), which enables reasoning directly in a joint latent space rather than in natural language. In MCOUT, the reasoning state is represented as a continuous hidden vector, iteratively refined and aligned with visual and textual embeddings, inspired by human reflective cognition. We develop two variants: MCOUT-Base, which reuses the language model's last hidden state as the continuous thought for iterative reasoning, and MCOUT-Multi, which integrates multimodal latent attention to strengthen cross-modal alignment between visual and textual features. Experiments on benchmarks including MMMU, ScienceQA, and MMStar show that MCOUT consistently improves multimodal reasoning, yielding up to 8.23% accuracy gains over strong baselines and improving BLEU scores up to 8.27% across multiple-choice and open-ended tasks. These findings highlight latent continuous reasoning as a promising direction for advancing LMMs beyond language-bound CoT, offering a scalable framework for human-like reflective multimodal inference.

## 1 Introduction

Vision-language models (VLMs) have transformed multimodal tasks, such as visual question answering (VQA), image captioning, and reasoning on benchmarks like ScienceQA (Lu et al., 2022), MMMU (Yue et al., 2024), and IQBench (Pham et al., 2025b), by seamlessly integrating visual and textual data. These models leverage visual models and large language models (LLMs) to process heterogeneous inputs, enabling applications from autonomous systems to interactive assistants. However, achieving robust reasoning in VLMs remains a challenge due to limitations in existing techniques, such as attention mechanism within the transformer decoder (Vaswani et al., 2017), prompting strategies like CoTs (Wei et al., 2022; Yao et al., 2023; Besta et al., 2024), or training methods like reinforcement learning (RL) (Ouyang et al., 2022; Pham & Ngo, 2025). CoT, originally developed for LLMs, prompts models to generate intermediate reasoning steps in natural language, while other approaches, such as fine-tuning with visual-text alignment, aim to enhance multimodal reasoning. These methods often rely on discrete token sequences or static vision features, leading to computational inefficiencies and difficulties in dynamically aligning visual and textual modalities for coherent reasoning, particularly in tasks requiring fine-grained multimodal understanding.

Inspired by human cognition, where reasoning involves generating intermediate thoughts and iteratively validating them against input data, such as revisiting images or documents to ensure coherence, we propose

the Multimodal Chain of Continuous Thought (MCOUT), a novel framework for efficient reasoning in VLMs. MCOUT operates in a unified latent space, dynamically aligning visual and textual representations to mimic reflective human thinking. We introduce two variants: MCOUT-Base, which uses the language model's last hidden state as a continuous thought for iterative refinement inspired from COCONUT (Hao et al., 2024), and MCOUT-Multi, which enhances cross-modal alignment by combining the hidden state with input embeddings via a multimodal latent attention mechanism. By overcoming the limitations of token-based CoT and static vision features, MCOUT reduces computational overhead by directly feeding hidden layers, with/without input embeddings, into the model as continuous thoughts. Tested on diverse benchmarks, MCOUT achieves significant performance gains, positioning it as a pioneering advancement in vision-language reasoning and offering a scalable approach for robust multimodal inference.

## 2    LITERATURE REVIEW

The development of reasoning capabilities in VLMs is critical for tasks like VQA and multimodal reasoning. Over the past few years, various techniques have been explored to enhance reasoning in both LLMs and VLMs, including attention mechanism, prompting techniques, training methods, and recent latent reasoning paradigms. CoT prompting, introduced by Wei et al. (2022), has significantly improved LLM performance on arithmetic tasks (e.g., GSM8K) and logical reasoning tasks (e.g., AQUA-RAT) by generating explicit intermediate steps. Building on CoT, its variants have emerged to address reasoning limitations. Self-consistency (Wang et al., 2022) samples multiple CoT outputs and selects the most consistent answer via majority voting, enhancing robustness but increasing computational cost. Tree of Thoughts (ToT) (Yao et al., 2023) structures reasoning as a tree search, exploring multiple paths for complex problem-solving, though its token-based nature remains computationally intensive. Graph of Thoughts (GoT) (Besta et al., 2024) extends ToT by modeling reasoning as a graph, enabling dynamic recombination of thoughts for greater efficiency. In the VLM domain, Multimodal CoT (Zhang et al., 2023) generates interleaved text and image reasoning steps, improving performance on ScienceQA but struggling with cross-modal alignment due to reliance on static vision features and verbose token sequences. These CoT methods, while effective for LLMs, face challenges in VLMs, where aligning heterogeneous modalities and minimizing token overhead are critical.

Beyond prompting, training techniques have been pivotal in enhancing reasoning for both LLMs and VLMs. RL methods, such as those explored by Ouyang et al. (2022), optimize LLMs using human feedback to improve reasoning and alignment, as seen in models like InstructGPT. Group relative policy Optimization (GRPO) (Shao et al., 2024) refines model outputs by incorporating reward signals, enhancing performance on complex tasks. Reasoning functions, such as RARL (Pham & Ngo, 2025), enable models to learn structured reasoning patterns through optimization, improving logical consistency. RL-based fine-tuning (Shen et al., 2025) has been applied to align visual and textual features, though these methods often rely on static embeddings, limiting dynamic reasoning capabilities. These training techniques complement prompting but still face challenges in efficiently integrating multimodal data for coherent reasoning.

To overcome these limitations, latent reasoning paradigms have shifted reasoning from discrete token sequences to continuous latent spaces. COCONUT (Hao et al., 2024) leverages the last hidden state of an LLM as a "continuous thought," enabling parallel exploration of reasoning paths via breadth-first search. This approach reduces token overhead and outperforms CoT on tasks requiring backtracking. Other latent reasoning methods for LLMs include Latent Reasoning Skills (LaRS) (Xu et al., 2023), which uses unsupervised learning to create latent representations of rationales, selecting in-context learning examples based on reasoning skills and achieving fourfold faster processing than CoT. Similarly, Wang et al. (2025) proposed a recurrent depth approach that iteratively refines latent representations, scaling test-time computation to enhance performance on complex tasks. These methods demonstrate the efficiency of latent reasoning in LLMs but are primarily designed for text-only contexts, leaving their application to VLMs largely unexplored.

In the VLM domain, latent reasoning is an emerging area with promising developments. Zhang et al. (2023) introduced a multimodal CoT framework that uses diffusion processes to learn a text-image aligned latent space, generating dynamic image features that improve reasoning on ScienceQA and multimodal machine translation. Yang et al. (2025a) developed MMaDA, a diffusion-based VLM that operates in latent spaces for coherent generation and reasoning across text and images, achieving strong performance in tasks like VQA and image captioning. Yang et al. (2025b) proposed the Mirage framework, which augments VLMs with latent visual tokens during decoding, enhancing reasoning efficiency in complex multimodal tasks. Fan & Zhou (2018) introduced stacked latent attention, preserving spatial information in latent spaces to improve reasoning in VQA tasks. Recent efforts, such as multimodal latent language modeling (Sun et al., 2024), employ next-token diffusion for continuous reasoning, while Corvid (Jiang et al., 2025) and Grounded Chain-of-Thought (GCoT) (Wu et al., 2025) address visual hallucination and decision-making accuracy. Despite these advances, most approaches rely on discrete token-based reasoning or static vision features, limiting efficient cross-modal alignment.

Inspired by human cognition and previous work (Hao et al., 2024), where reasoning involves generating intermediate thoughts and iteratively validating them against input data, our MCOUT addresses these gaps by introducing a novel latent reasoning framework for VLMs, specifically for image-based tasks. MCOUT employs two variants: MCOUT-Base, which uses the language model's last hidden state as a continuous thought for iterative refinement, and MCOUT-Multi, which integrates the hidden state with image embeddings via a multimodal latent attention mechanism, enabling dynamic alignment of visual and textual representations. MCOUT mimics human reflective reasoning by iteratively refining thoughts in a continuous latent space, as demonstrated in our implementation, which supports multimodal inputs and has been tested successfully for vision-language reasoning. MCOUT offers a significant advancement, bridging the efficiency of latent reasoning with the complexity of vision-language reasoning, paving the way for robust and scalable VLMs.

## 3 METHODOLOGY

### 3.1 MODEL ARCHITECTURE

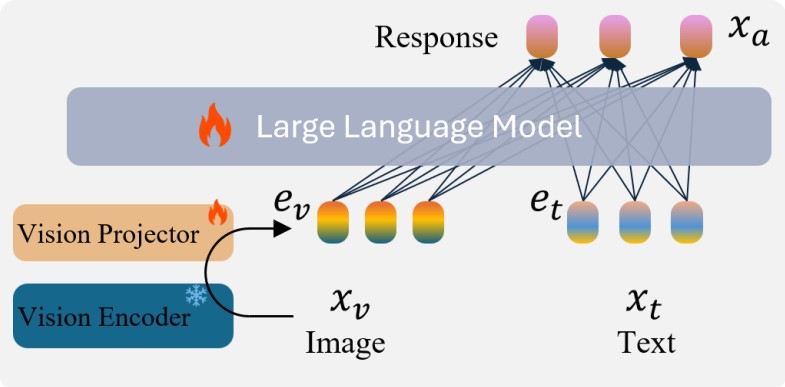

Figure 1: Model architecture.

The MCOUT framework is built upon a vision-language model, SilVar (Pham et al., 2025a), comprising a pre-trained visual encoder $\mathcal{V}$ and a language model $\mathcal{L}$, as illustrated in Figure 1. We use CLIP (Radford et al., 2021) as the visual encoder $\mathcal{V}$, which processes input images $\mathbf{x}_v \in \mathbb{R}^{H \times W \times C}$ to produce visual embeddings $\mathbf{e}_v \in \mathbb{R}^{S_v \times D}$, where $S_v$ is the sequence length of visual tokens and $D$ is the embedding dimension. For the

language model $\mathcal{L}$, we employ Llama 3.2 1B, which processes tokenized text inputs $\mathbf{x}_t$ to generate contextual embeddings $\mathbf{e}_t \in \mathbb{R}^{S_t \times D}$, where $S_t$ is the sequence length of text tokens. In this study, we use CLIP and Llama 3.2 1B for all experiments because we want to focus on latent reasoning for small VLMs, although our pipeline is compatible with other LLMs.

For MCOUT-Multi, the core component is the multimodal latent attention module, which integrates the language model's last hidden state $\mathbf{h}_l \in \mathbb{R}^{B \times D}$ for a batch of $B$ samples with multimodal input embeddings $\mathbf{e}_m \in \mathbb{R}^{B \times S_m \times D}$ (for images, $\mathbf{e}_m = \mathbf{e}_v$). The module projects $\mathbf{h}_l$ into a query space, applies multi-head attention with $N_h = 8$ heads to attend to $\mathbf{e}_m$, and normalizes the output to produce a thought embedding:

$$\mathbf{h}_t = \text{Norm}(\text{Proj}_{\text{back}}(\text{MultiHeadAttn}(\text{Proj}(\mathbf{h}_l), \mathbf{e}_m^\top))) \in \mathbb{R}^{B \times 1 \times D}, \tag{1}$$

where $\text{Proj} : \mathbb{R}^D \to \mathbb{R}^D$ and $\text{Proj}_{\text{back}} : \mathbb{R}^D \to \mathbb{R}^D$ are linear projections, and Norm denotes layer normalization. This process enriches $\mathbf{h}_t$ with visual context for cross-modal alignment. In contrast, MCOUT-Base bypasses this module, directly using the last hidden state as the thought embedding:

$$\mathbf{h}_t = \mathbf{h}_l \in \mathbb{R}^{B \times 1 \times D}. \tag{2}$$

MCOUT-Base relies on the language model's internal state for reasoning, while MCOUT-Multi enhances it through multimodal fusion, mimicking human reflective reasoning by validating thoughts against input embeddings.

## 3.2 MULTIMODAL LATENT REASONING

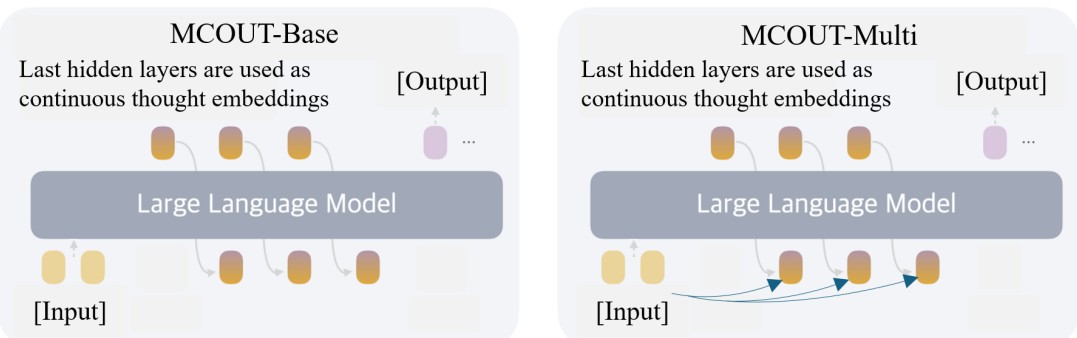

Figure 2: Comparison between two Chain of Continuous Thought approaches: MCOUT-Base (left) vs. MCOUT-Multi (right).

The MCOUT framework performs reasoning by iteratively generating continuous thought representations in a latent space, inspired by human cognition, where intermediate thoughts are validated against input data for coherence, as shown in Figure 2. Given preprocessed interleaved input embeddings $\mathbf{e}_{\text{inter}} \in \mathbb{R}^{B \times S_{\text{max}} \times D}$ and an attention mask $\mathbf{m} \in \{0, 1\}^{B \times S_{\text{max}}}$ for a batch of $B$ samples with maximum sequence length $S_{\text{max}}$, the language model $\mathcal{L}$ computes hidden states:

$$\mathbf{h} = \mathcal{L}(\mathbf{e}_{\text{inter}}, \mathbf{m}) \in \mathbb{R}^{B \times S_{\text{max}} \times D}. \tag{3}$$

The last hidden state for each sample is extracted by selecting the hidden state corresponding to the last non-padded token:

$$\mathbf{h}_l = \mathbf{h}[\cdot, \text{argmax}(\mathbf{m}, \text{dim} = 1) - 1, \cdot] \in \mathbb{R}^{B \times D}. \tag{4}$$

For $N_t$ latent reasoning steps, MCOUT iteratively produces thought embeddings $\mathbf{h}_t^{(k)}$ for $k = 1, \ldots, N_t$. As mentioned, we explore two approaches: MCOUT-Base directly feeds the last hidden state to the language model $N_t$ times, while MCOUT-Multi combines the last hidden state with input embeddings before feeding the resulting thought embedding to the language model:

- In MCOUT-Base:

$$\mathbf{h}_t^{(k)} = \mathbf{h}_l^{(k-1)} \in \mathbb{R}^{B \times 1 \times D},$$ (5)

- In MCOUT-Multi:

$$\mathbf{h}_t^{(k)} = \text{MultimodalLatentAttention}(\mathbf{h}_l^{(k-1)}, \mathbf{e}_m) \in \mathbb{R}^{B \times 1 \times D}.$$ (6)

Each thought embedding is appended to the input sequence:

$$\mathbf{e}_{\text{inter}}^{(k)} = [\mathbf{e}_{\text{inter}}^{(k-1)}, \mathbf{h}_t^{(k)}] \in \mathbb{R}^{B \times (S_{\max}+k) \times D},$$ (7)

$$\mathbf{m}^{(k)} = [\mathbf{m}^{(k-1)}, \mathbf{1}_{B \times 1}] \in \{0,1\}^{B \times (S_{\max}+k)}.$$ (8)

The updated sequence is fed back into the language model to compute the next hidden state, repeating for $N_t$ iterations. In the final step ($k = N_t + 1$), the language model generates the output sequence ($\mathbf{x}_a$) using a standard generation process. The loss function for training combines an auxiliary loss for intermediate thoughts (weighted by $\mu$) and the final output loss:

$$\mathcal{L}_{\text{total}} = \sum_{k=1}^{N_t} \mu \cdot \mathcal{L}_{\text{aux}}^{(k)} + \mathcal{L}_{\text{final}},$$ (9)

where $\mathcal{L}_{\text{aux}}^{(k)}$ is the language modeling loss for the $k$-th thought, and $\mathcal{L}_{\text{final}}$ is the loss for the final output, computed using cross-entropy over the target tokens.

## 4 EXPERIMENT AND RESULT

### 4.1 DATASETS AND TRAINING

To evaluate the effectiveness of our MCOUT framework, we conducted experiments using four diverse vision-language datasets: VQAv2 (Goyal et al., 2017), MMMU (Yue et al., 2024), ScienceQA (Lu et al., 2022), and MMStar (Chen et al., 2024). These datasets assess the model's reasoning capabilities across multimodal tasks, including VQA, scientific reasoning, and general knowledge understanding, with a focus on image-text integration. The VQAv2 dataset, used for pretraining, contains 443,757 question-answer pairs associated with images from the COCO dataset, emphasizing tasks like object recognition, attribute identification, and spatial reasoning.

The MMMU dataset, employed for fine-tuning, includes approximately 150 training samples and 900 validation samples. We also utilize the ScienceQA dataset, which focuses on scientific reasoning across natural science, social science, and language science. For this dataset, we use a subset of 6,218 training samples that contain both text and image contexts. The subset was chosen to preserve modality and format distributions while enabling fair ablations (MCOUT-Base/MCOUT-Multi, $N_t$, and $\mu$) within a single-GPU training. The MMStar dataset, used exclusively for testing, consists of 1,500 test samples with curated image-question-answer triplets, designed for challenging visual reasoning tasks like object counting and scene understanding. All datasets are preprocessed to ensure compatibility with MCOUT's image-based pipeline, with images resized to $224 \times 224$ pixels and text tokenized to a maximum context length of 1024 tokens, interleaved with visual embeddings for unified input processing.

For training, we develop a multimodal model as described in Section 3.1, consisting of a pre-trained CLIP vision encoder and a Llama 3.2 1B language model. We pretrained the model on the VQAv2 training dataset for 1 epoch, followed by fine-tuning on ScienceQA and MMMU for 10 epochs. The model employs 8-bit precision, freezes the vision model, and uses LoRA (rank 64, alpha 16) for efficient adaptation. Training is conducted on a single CUDA device with 2 compute workers, using a batch size of 4 and a linear warmup cosine learning rate schedule (initial LR: $1 \times 10^{-5}$, minimum LR: $1 \times 10^{-6}$, warmup LR: $1 \times 10^{-6}$, weight decay: 0.05). The number of latent thoughts is experimented with values of 5 and 10 for both MCOUT-Base and MCOUT-Multi approaches, enabling iterative reasoning in a continuous latent space. During inference, we set the temperature to 0.1 for all experiments.

## 4.2 RESULTS AND BENCHMARKING

Table 1: Performance on the ScienceQA test set.

| Models | Parameters (B) | accuracy (%) | BLEU |
|---|---|---|---|
| *Our experiments* | | | |
| Baseline | 1 | 56.17 | 51.48 |
| MCOUT-Base ($N_t = 5$) | 1 | 58.60 (↑ 4.33%) | 52.44 (↑ 1.87%) |
| MCOUT-Multi ($N_t = 5$) | 1 | 58.45 (↑ 4.05%) | **52.60 (↑ 2.18%)** |
| MCOUT-Base ($N_t = 10$) | 1 | **58.86 (↑ 4.79%)** | 52.31 (↑ 1.61%) |
| MCOUT-Multi ($N_t = 10$) | 1 | 58.20 (↑ 3.61%) | 52.27 (↑ 1.53%) |
| *Literature reports* | | | |
| Kosmos2 (Peng et al., 2023) | 1.7 | 32.70 | – |
| SilVar (Pham et al., 2025a) | 7 | 63.21 | – |
| LLaVA-7B (Liu et al., 2023) | 7 | 41.10 | – |
| InstructBLIP-7B (Dai et al., 2023) | 8 | 54.10 | – |
| OpenFlamingo (Awadalla et al., 2023) | 9 | 44.80 | – |
| Qwen-VL (Bai et al., 2023) | 9.6 | 61.10 | – |
| MiniGPT-4 (Zhu et al., 2023) | 13 | 47.71 | – |
| LLaMA2-13B (Yang et al., 2023) | 13 | 55.78 | – |
| LLaVA-13B (Yang et al., 2023) | 13 | 47.74 | – |
| PandaGPT-13B (Su et al., 2023) | 13 | 63.20 | – |

To evaluate the MCOUT framework, we compare MCOUT-Base and MCOUT-Multi against our baseline VLM without latent reasoning. Evaluations are conducted on the ScienceQA and MMMU validation sets and the MMStar test set, using accuracy and BLEU. We also compare our small VLM with other models. Tables 1, 2, and 3 summarize the results of our models on the ScienceQA, MMMU validation and MMStart benchmark, respectively.

For ScienceQA, as shown in Table 1, MCOUT-Base ($N_t = 10$) achieves the highest accuracy at 58.86% (up 4.79%), while MCOUT-Multi ($N_t = 5$) leads in BLEU at 52.60 (up 2.18%), excelling in image-heavy scientific reasoning due to its multimodal attention mechanism. With 1B parameters, both variants outperform larger models like Kosmos-2 (1.7B, 32.70%), LLaVA-7B/13B (41.10%–47.74%), and MiniGPT-4-13B (47.71%), and closely match InstructBLIP-7B (8B, 54.10%) and LLaMA-2-13B (55.78%), showcasing MCOUT's efficiency in leveraging iterative reasoning for robust performance.

For MMMU, as illustrated in Table 2, MCOUT-Base ($N_t = 5$) achieves the highest gains, with accuracy at 27.53% (up 8.21%) and BLEU at 27.54 (up 8.31%). MCOUT-Multi ($N_t = 10$) follows closely with 7.54% and 7.58% gains in accuracy and BLEU, respectively, leveraging multimodal attention for cross-modal

Table 2: Performance on the MMMU validation set.

| Models | Parameters (B) | accuracy (%) | BLEU |
|---|---|---|---|
| *Our experiments* | | | |
| Baseline | 1 | 25.44 | 25.44 |
| MCOUT-Base ($N_t = 5$) | 1 | **27.53** (↑ **8.21%**) | **27.54** (↑ **8.31%**) |
| MCOUT-Multi ($N_t = 5$) | 1 | 27.18 (↑ 6.79%) | 27.19 (↑ 6.82%) |
| MCOUT-Base ($N_t = 10$) | 1 | 27.52 (↑ 8.18%) | **27.54** (↑ **8.31%**) |
| MCOUT-Multi ($N_t = 10$) | 1 | 27.36 (↑ 7.54%) | 27.37 (↑ 7.58%) |
| *Literature reports* | | | |
| Kosmos 2 (Peng et al., 2023) | 1.7 | 23.7 | – |
| MiniGPT-4-v1-7B (Zhu et al., 2023) | 7 | 23.6 | – |
| LLaVA-v1.5-7B (Liu et al., 2023) | 7 | 33.7 | – |
| MiniGPT-4-v2 (Chen et al., 2023) | 7 | 25.0 | – |
| OpenFlamingo v2 (Awadalla et al., 2023) | 9 | 28.8 | – |
| Qwen-VL (Bai et al., 2023) | 9.6 | 29.6 | – |
| LLaVA-v1.5-13B (Liu et al., 2023) | 13 | 37.0 | – |
| PandaGPT-13B (Su et al., 2023) | 13 | 32.9 | – |

tasks. With 1B parameters, MCOUT outperforms Kosmos-2 and MiniGPT-4 variants, and nearly matches OpenFlamingo-9B and Qwen-VL, demonstrating strong efficiency in college-level reasoning.

Table 3: Performance on the MMStar test set.

| Models | Parameters (B) | accuracy (%) | BLEU |
|---|---|---|---|
| *Our experiments* | | | |
| Baseline | 1 | 25.13 | 25.14 |
| MCOUT-Base ($N_t = 10$) | 1 | **26.13** (↑ **3.98%**) | **26.14** (↑ **3.98%**) |
| MCOUT-Multi ($N_t = 10$) | 1 | 26.07 (↑ 3.74%) | 26.08 (↑ 3.74%) |
| *Literature reports* | | | |
| Kosmos2 (Peng et al., 2023) | 1.7 | 24.9 | – |
| MiniGPT-4-v1-7B (Zhu et al., 2023) | 7 | 16.3 | – |
| MiniGPT-4-v2 (Chen et al., 2023) | 7 | 21.3 | – |
| LLaVA-7B (Liu et al., 2023) | 7 | 27.1 | – |
| OpenFlamingo v2 (Awadalla et al., 2023) | 9 | 26.9 | – |
| Qwen-VL-Chat (Bai et al., 2023) | 9.6 | 34.5 | – |
| PandaGPT-13B (Su et al., 2023) | 13 | 25.6 | – |

For MMStar, as illustrated in Table 3, MCOUT-Base ($N_t = 10$) improves accuracy and BLEU by 3.98%, while MCOUT-Multi ($N_t = 10$) gains 3.74% in both metrics, enhancing fine-grained visual reasoning through iterative thought generation. Despite its 1B parameters, MCOUT outperforms Kosmos-2, MiniGPT-4-v1-7B, MiniGPT-4-v2, and PandaGPT-13B, and closely rivals OpenFlamingo-9B and LLaVA-7B, highlighting its efficiency in challenging visual tasks.

### 4.3 MULTIMODAL LATENT REASONING ANALYSIS

To understand the performance differences and similarities between MCOUT-Base and MCOUT-Multi, we analyzed their latent distributions, as illustrated in Figure 3. Prior to training, we identified a significant norm imbalance: the last hidden state norm was **103.90**, while the initial thought embedding norm (from multimodal attention) was **26.48** on ScienceQA, posing a risk of unstable fusion in MCOUT-Multi. To

mitigate this, we introduced normalization layers with a final normalization step-into the attention module (Equation 1), aligning the scales and stabilizing the thought embeddings. For MCOUT-Base, which uses the last hidden state directly ($\mathbf{h}_t = \mathbf{h}_l$), the mean of the last hidden layer starts at -0.02197 and fluctuates slightly with a consistent standard deviation of approximately 2.23, as shown in the top figures, reflecting a stable reasoning process that underpins its performance gains (4.79% accuracy improvement on ScienceQA and 8.21% on MMMU).

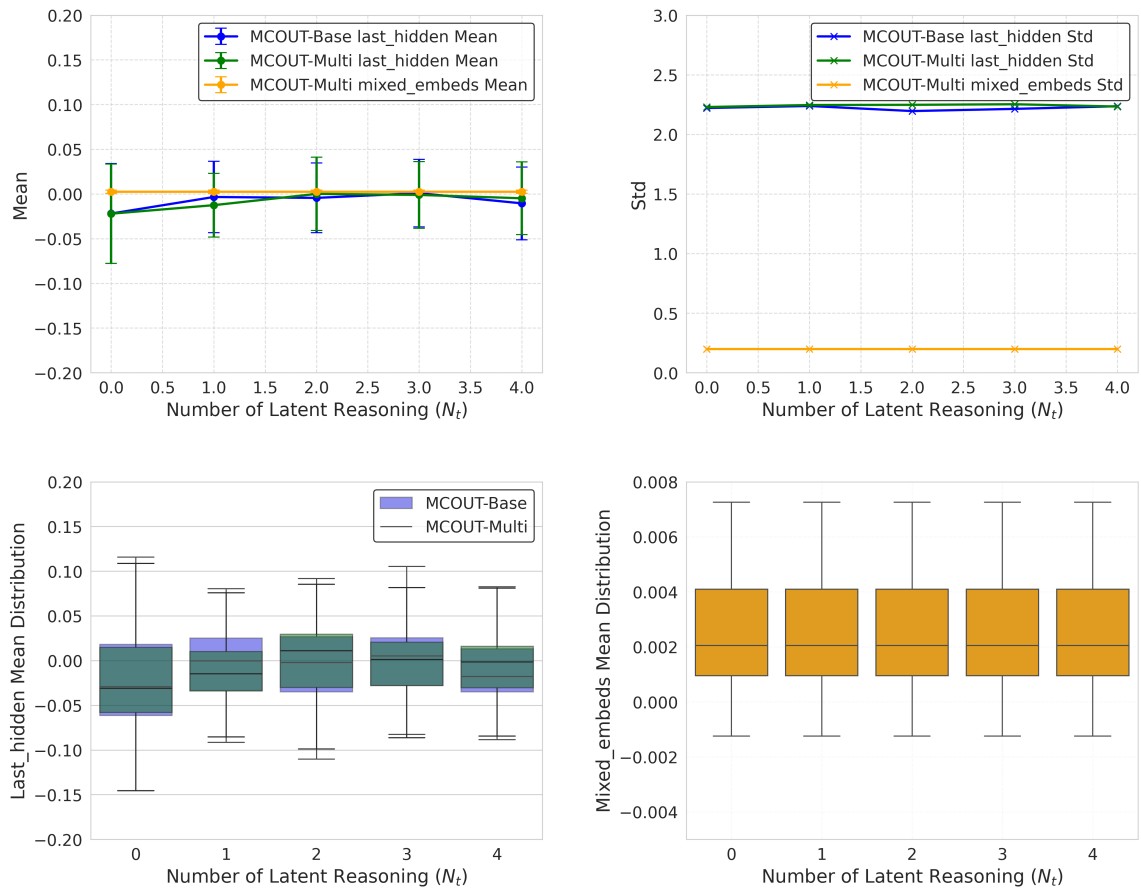

Figure 3: Latent distribution analysis of MCOUT-Base and MCOUT-Multi, showing mean and standard deviation of last hidden states and mixed embeddings across 100 samples and 5 thought iterations.

MCOUT-Multi, which integrates the last hidden state with multimodal input embeddings, shows last hidden layer means ranging from -0.02212 to -0.00112 across iterations, with standard deviations around 2.24, closely tracking MCOUT-Base's patterns and indicating minimal disruption from multimodal fusion. However, the mixed embeddings reveal a critical limitation: their mean remains constant at 0.002418 with a negligible standard deviation of 0.001866, and the mixed standard deviation is uniformly 0.198925 across all iterations, suggesting a static, low-variance contribution from the multimodal input. This persistent uniformity, despite normalization, points to a modality collapse, where the attention mechanism fails to extract diverse visual context, aligning MCOUT-Multi's performance (58.45% accuracy at $N_t = 5$) closely with MCOUT-Base (58.60%). This observation resonates with the sinking of visual attention in recent studies (Kang et al., 2025;

Cancedda, 2024; Sim et al., 2025), which attributes such collapse to activation imbalances favoring a specific type (e.g. text). Our study shows that low-variance embeddings (mixed_embeds std $\approx 0.2$ vs. last_hidded std $\approx 2.2$) limit multimodal benefits. The pre-training norm adjustment likely prevented catastrophic fusion failure, but the static mixed embeddings suggest entropy collapse (Zhai et al., 2023), where uniform attention weights diminish multimodal impact.

## 5 ABLATION STUDY

To investigate the impact of the auxiliary weight $\mu$ in the MCOUT loss function (Equation 9), we conduct an ablation study with the impact of the auxiliary weight $\mu$ in the MCOUT loss function with $N_t = 5$, as shown in Table 4. $\mu = 0.3$ yields the highest performance, improving ScienceQA accuracy by 4.33%, and MMMU accuracy by 8.23%, highlighting the importance of balancing auxiliary thought supervision for effective multimodal reasoning. Higher $\mu$ values (0.5, 0.8) reduce gains, suggesting overemphasis on intermediate thoughts may disrupt final output optimization, while $\mu = 0$ yields moderate improvements. Although using an auxiliary loss boosts model performance, it increases training time based on our experiments.

We also evaluate fully finetuning both the vision encoder and language model with LoRA. For MCOUT, we use $N_t = 5$ and $\mu = 0$. As shown in Table 4, finetuning boosts performance further, with improvements ranging from 3.06% to 5.88% across benchmarks. The performance gap between MCOUT-Base and MCOUT-Multi remains small, indicating that both strategies benefit consistently from full finetuning. These results reinforce the effectiveness of our method and demonstrate that MCOUT's iterative reasoning remains robust under different optimization settings, confirming the stability and adaptability of our framework.

Table 4: Ablation study for ScienceQA test and MMMU val using $N_t = 5$.

| Models | Auxiliary weight ($\mu$) | ScienceQA test | | MMMU val | |
|---|---|---|---|---|---|
| | | accuracy | BLEU | accuracy | BLEU |
| Baseline | | 56.17 | 51.48 | 25.44 | 25.44 |
| MCOUT-Base | 0 | 58.12 (↑ 3.47%) | 52.05 (↑ 1.11%) | 27.41 (↑ 7.75%) | 27.43 (↑ 7.82%) |
| MCOUT-Base | 0.3 | **58.60 (↑ 4.33%)** | **52.44 (↑ 1.87%)** | **27.53 (↑ 8.23%)** | **27.54 (↑ 8.27%)** |
| MCOUT-Base | 0.5 | 57.56 (↑ 2.48%) | 52.10 (↑ 1.20%) | 26.44 (↑ 3.93%) | 26.44 (↑ 3.93%) |
| MCOUT-Base | 0.8 | 57.52 (↑ 2.40%) | 52.00 (↑ 1.01%) | 25.90 (↑ 1.81%) | 25.91 (↑ 1.85%) |
| *Fully finetuning model with LoRA* | | | | | |
| Baseline | | 62.61 | 54.96 | 26.55 | 26.56 |
| MCOUT-Base | 0 | 64.60 (↑ 3.18%) | **56.73 (↑ 3.22%)** | 27.98 (↑ 5.39%) | 27.99 (↑ 5.39%) |
| MCOUT-Multi | 0 | **64.75 (↑ 3.42%)** | 56.64 (↑ 3.06%) | **28.11 (↑ 5.88%)** | **28.11 (↑ 5.83%)** |

## 6 CONCLUSION

In this work, we investigated multimodal reasoning for a small VLM through two key contributions: (1) building a 1B-parameter vision-language model, and (2) proposing the Multimodal Chain of Continuous Thought (MCOUT) framework, which employs a step-by-step reasoning process inspired by human reflection. MCOUT improves performance, achieving gains of up to 8.23% in accuracy on MMMU and 4.79% on ScienceQA. As a pioneering effort to explore multimodal continuous latent reasoning, our study provides a promising foundation for efficient multimodal reasoning. Despite these advances, aligning input embeddings with the final hidden layers remains a challenge, as it complicates multimodal alignment in MCOUT and increases training time. Going forward, we will investigate multimodal attention and alternative methods for multimodal alignment within continuous latent reasoning.

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
