# OpenReview forum: "Multimodal Chain of Continuous Thought for Latent-Space Reasoning in Vision-Language Models"
_ICLR.cc/2026/Conference — ICLR 2026 Conference Withdrawn Submission_

### Official Review · Reviewer_kxzQ · 2025-10-21

**Soundness:** 2
**Presentation:** 1
**Contribution:** 1
**Rating:** 2
**Confidence:** 5

**Summary:**

The paper propose MCOUT, a method to incoporate model hidden representation iteratively to simulate the process of chain-of-thought in the latent space. The paper additionally proposes MCOUT-multi to use an attention module to merge the visual embeddings and the intermediate hidden representations. Evaluation on vision-language benchmarks demonstrate the advantage of the intermediate latent reasoning compared to the baseline model without it.

**Strengths:**

N/A

**Weaknesses:**

* The comparison of results is very unclear. For example, MCOUT-Base (Nt = 10) is written to have "up 4.79%" with 58.86% accuracy, while the baseline has 56.17. It is unclear what the method is comparing against. In addition the comparison with other baseline models is unfair because all the MCOUT models are finetuned on the benchmarks.

* It is unclear what the purpose of the Multimodal Latent Attention module is. The discussion in Section 4.3 introduces many statistical results but are largely unrelated in justifying the module. Based on the experimental results, it seems that the MCOUT-Base consistently outperforms MCOUT-Multi.

* The Training and evaluation setting is strange. It used only VQAv2 for pretraining as opposed to the standard LLaVA-style 2 stage training. Then, the model is finetuned only on the downstream benchmarks only. This raises question on the scalability and generalizability of the approach. In addition, the method doesn't include any CoT data during training, making it unclear how or whether the model can gain CoT capabilities at all.

**Questions:**

N/A

---

### Official Review · Reviewer_YEkc · 2025-10-28

**Soundness:** 3
**Presentation:** 3
**Contribution:** 3
**Rating:** 6
**Confidence:** 4

**Summary:**

This paper introduces a novel reasoning framework for Vision-Language Models (VLMs), termed "Multimodal Chain of Continuous Thought" (MCOUT). Its core contribution lies in shifting the reasoning process from discrete natural language sequences to iterative refinements within a continuous latent space. This represents a significant, exploratory breakthrough from the traditional "Chain-of-Thought" (CoT) paradigm, aiming to address inefficiencies and insufficient multimodal alignment that arise when forcing heterogeneous modalities into a single language sequence in multimodal contexts.

**Strengths:**

1. The paper introduces a novel paradigm for vision language model reasoning, shifting from discrete natural language sequences to iterative refinements within a continuous latent space. This approach effectively challenges the inherent limitations of language centric Chain of Thought methods, presenting a promising avenue for more intrinsic and efficient reasoning.

2. MCOUT achieves substantial performance improvements on several multimodal benchmarks using only a one billion parameter vision language model. These results indicate significant efficiency, allowing smaller models to attain powerful reasoning capabilities that outperform larger baselines, which is particularly valuable for resource constrained environments.

3. The paper's rigorous analysis, especially the latent space dynamics, provides critical diagnostic insights into the current limitations of multimodal fusion mechanisms. By empirically revealing issues like modality collapse, the study not only identifies a key challenge for the field but also offers a clear, interpretable, and readily extendable foundation for further research into advanced reasoning.

**Weaknesses:**

1. The paper proposes multimodal latent attention to enhance cross-modal alignment. However, the conducted latent space analysis reveals that the mixed embeddings exhibit remarkably low variance across iterations. This observation strongly suggests that the multimodal attention mechanism fails to extract and integrate discriminative visual context effectively, leading to MCOUT-Multi’s performance being almost indistinguishable from MCOUT-Base. Consequently, the claimed advantage in dynamic multimodal alignment is not robustly supported by empirical evidence.

2. While the authors identify and acknowledge the issue of modality collapse, particularly evidenced by the static nature of mixed embeddings and initial norm imbalances, the proposed solutions, such as normalization layers, appear to be primarily engineering fixes rather than fundamental architectural or algorithmic advancements. A more profound exploration and resolution of this critical weakness in dynamic multimodal integration are absent.

3. The ablation studies demonstrate that the framework's performance is notably sensitive to the number of latent reasoning steps (Nt) and the auxiliary loss weight (µ). Achieving optimal results necessitates careful tuning of these hyperparameters, which implies a potential lack of robustness or increased deployment complexity in varied application scenarios.

4. The paper posits improved efficiency by circumventing token generation. Nevertheless, a detailed quantitative analysis comparing the incremental computational overhead of multiple forward passes inherent in the iterative process against the costs associated with traditional discrete token generation (CoT) is not provided. This omission leaves the actual efficiency gains less substantiated.

**Questions:**

1. Did the authors investigate alternative attention mechanisms or different integration points for visual information to mitigate the observed modality collapse in MCOUT-Multi?
2. Could the authors provide a more detailed comparative analysis of MCOUT's inference time and memory footprint against traditional Chain of Thought methods, especially across varying input lengths and iteration counts?
3. Was a comprehensive error analysis performed to identify specific types of multimodal reasoning tasks where MCOUT underperforms, potentially with illustrative failure examples to elucidate underlying limitations?

---

### Official Review · Reviewer_BxVz · 2025-10-30

**Soundness:** 2
**Presentation:** 2
**Contribution:** 2
**Rating:** 2
**Confidence:** 5

**Summary:**

This paper proposes the Multimodal Chain of Continuous Thought (MCOUT), a framework for vision-language models that enables reasoning directly in a joint latent space (instead of discrete natural language CoT) by representing "thought" as an iteratively refined continuous hidden vector, mimicking human reflective cognition. It presents two variants: 1) MCOUT-Base, which reuses the model’s last hidden state; and 2) MCOUT-Multi, which integrates multimodal latent attention to enhance cross-modal alignment. Experiments are conducted on benchmarks including MMMU and ScienceQA using the Llama3.2 1B model, with MCOUT achieving up to 8.23% accuracy improvement over baselines.

**Strengths:**

The work makes a pioneering effort to shift reasoning for multimodal tasks from discrete natural language (traditional CoT) to a continuous latent space, representing an innovative exploration of latent-space reasoning in multimodal scenarios.

**Weaknesses:**

1.	Limited Novelty and Inadequate Comparison
The core method largely transfers latent CoT from COCONUT to a multimodal context, lacking sufficient novelty. Additionally, it fails to adequately compare against relevant existing methods (e.g., Mirage) and omits key baselines (e.g., COCONUT adapted to multimodal scenarios), making its distinct advantages unclear.

2.	Ambiguity in Method and Presentation

The paper lacks clarity: the introduction fails to clearly articulate specific contributions, and the claim of "improved efficiency" (cited as a motivation) is not experimentally validated.

Methodological details are unclear, including the training process, calculation of L_{aux}, and the rationale for introducing MCOUT-Base and MCOUT-Multi.

3.	Flaws in Experimental Design

The experimental setup is ambiguous: baselines for MCOUT are not clearly defined, and the inclusion of scores from unrelated models is confusing (serving no clear comparative purpose).

The performance gain of MCOUT-Multi over MCOUT-Base is marginal, with no in-depth analysis of this phenomenon to justify MCOUT-Multi’s value.

Experiments are limited to a single small model (Llama3.2 1B), lacking validation across multiple model sizes or series.

**Questions:**

1.	How is the auxiliary loss L_{aux} calculated?

2.	How are the parameters of the MultiHeadAttn module initialized?

3.	Given the insignificant performance difference between MCOUT-Base and MCOUT-Multi, what is the rationale for introducing these two settings? Further, why is Section 4.3 significant under this circumstance?

4.	In Table 4, when the auxiliary weight is 0, performance is comparable to the best setting (weight = 0.3). Does this imply L_{aux} is useless? And why is performance with an auxiliary weight of 0 much higher than the baseline?

---

### Official Review · Reviewer_yJPg · 2025-11-05

**Soundness:** 2
**Presentation:** 3
**Contribution:** 2
**Rating:** 2
**Confidence:** 4

**Summary:**

This paper proposes MCOUT (Multimodal Chain of Continuous Thought), a framework for reasoning in vision-language models (VLMs) that operates in continuous latent space rather than generating discrete reasoning tokens. The method iteratively refines hidden state representations by feeding them back into the model as "thought tokens" over N_t iterations. Two variants are proposed: MCOUT-Base, which directly reuses the last hidden state, and MCOUT-Multi, which fuses the hidden state with visual embeddings via multimodal attention. Experiments on ScienceQA, MMMU, and MMStar show improvements of 3-8% accuracy over baselines using a 1B parameter model (CLIP + Llama 3.2 1B).

**Strengths:**

1. The paper articulates a compelling case for continuous latent reasoning over token-based CoT. By generating N_t=5-10 hidden state vectors instead of 50-100 reasoning tokens, the method achieves computational efficiency while maintaining interpretable iteration counts.

2. The paper is generally well written with a clear motivation, good writing, and informative figures.

3. Section 4.3's analysis of modality collapse in MCOUT-Multi is commendable. The authors identify low-variance mixed embeddings and connect this to recent work on attention sinking, providing genuine insight into when/why their multimodal fusion struggles.

**Weaknesses:**

Here's the review with unified, simpler formatting:

1. **MCOUT-Base is essentially COCONUT applied to VLMs**: The paper positions MCOUT-Base as a novel contribution, but it appears methodologically identical to COCONUT (Hao et al., 2024) - taking the last hidden state and feeding it back iteratively. The only difference is applying it to a VLM architecture (which is just an LLM with a vision encoder). This should be acknowledged more directly. The true novel contribution is MCOUT-Multi, but...

2. **MCOUT-Multi underperforms MCOUT-Base, undermining the paper's core thesis**: Unfortunately, the paper's actual novel contribution (multimodal latent attention) generally performs worse than or comparable to the baseline COCONUT (MCOUT-Base) approach. The paper essentially demonstrates that COCONUT works for VLMs, but that adding multimodal fusion doesn't help (and often hurts). This fundamentally undermines the premise that multimodal latent reasoning requires special cross-modal alignment.

   More critically, if the paper's contribution is showing that multimodal information isn't effectively utilized in continuous latent CoT, then the experimental investigation is insufficient. ScienceQA, MMMU, and MMStar suffer from significant language bias - many questions can be answered from text alone without visual grounding. This has been documented by the literature NaturalBench (Lin et. al.). If the datasets don't require deep visual integration, MCOUT-Multi's fusion mechanism may appear ineffective simply because it's not needed. The paper should evaluate on benchmarks that require visual grounding such as Winoground, NaturalBench, or BLINK.

   Without this investigation, we can't distinguish between: Hypothesis A - multimodal fusion in continuous latent space is fundamentally flawed; Hypothesis B - the specific fusion mechanism needs refinement; Hypothesis C - the datasets don't require multimodal reasoning, so fusion provides no benefit; or Hypothesis D - training was insufficient for the fusion module to learn effective alignment.

3. **Custom model limits generalizability**: The authors train their own 1B VLM from scratch rather than applying MCOUT to existing models (LLaVA, Qwen-VL, InstructBLIP, etc.). This is problematic because the method is model-agnostic and could be applied to any VLM without retraining, comparisons are unfair since their baseline is their own undertrained model rather than SOTA VLMs, we can't assess whether improvements come from the method or their specific training recipe, and the approach would be more convincing if it improved existing strong models.

4. **Limited baselines**: No comparison to standard text-based CoT prompting on the same model, self-consistency or other CoT variants, other test-time scaling methods, or the computational cost comparison (inference time, memory) versus token-based reasoning.

While the core idea and approach may have merit, it needs more distinction from prior work, more thorough baseline comparions, and better justification of the main motivation for multimodal fusion.

**Questions:**

Can you explicitly clarify the methodological differences between MCOUT-Base and COCONUT? If they are identical, should MCOUT-Base be framed as an application/extension of COCONUT rather than a separate contribution?

I would also appreciate if you can address the open questions left open in the Weaknesses section.

---

### Note · Authors · 2026-01-23

I have read and agree with the venue's withdrawal policy on behalf of myself and my co-authors.